# Prevalence of Carbapenemase and Extended-Spectrum β-Lactamase Producing *Enterobacteriaceae*: A Cross-Sectional Study

**DOI:** 10.3390/antibiotics12010148

**Published:** 2023-01-11

**Authors:** Muhammad Muqaddas Mustafai, Mavra Hafeez, Safa Munawar, Sakeenabi Basha, Ali A. Rabaan, Muhammad A. Halwani, Abdulsalam Alawfi, Amer Alshengeti, Mustafa A. Najim, Sara Alwarthan, Meshal K. AlFonaisan, Souad A. Almuthree, Mohammed Garout, Naveed Ahmed

**Affiliations:** 1Department of Medical Education, Quaid e Azam Medical College, Bahawalpur 63100, Pakistan; 2Department of Emergency, Tehsil Headquarter Hospital Mailsi, Vehari 61200, Pakistan; 3Department of Medical Education, King Edward Medical University, Lahore 54000, Pakistan; 4Department of Medical Education, Nawaz Sharif Medical College, Gujrat 50700, Pakistan; 5Department of Community Dentistry, Faculty of Dentistry, Taif University, Taif 21944, Saudi Arabia; 6Molecular Diagnostic Laboratory, Johns Hopkins Aramco Healthcare, Dhahran 31311, Saudi Arabia; 7College of Medicine, Alfaisal University, Riyadh 11533, Saudi Arabia; 8Department of Public Health and Nutrition, The University of Haripur, Haripur 22610, Pakistan; 9Department of Medical Microbiology, Faculty of Medicine, Al Baha University, Al Baha 4781, Saudi Arabia; 10Department of Pediatrics, College of Medicine, Taibah University, Al-Madinah 41491, Saudi Arabia; 11Department of Infection Prevention and Control, Prince Mohammad Bin Abdulaziz Hospital, National Guard Health Affairs, Al-Madinah 41491, Saudi Arabia; 12Department of Medical Laboratories Technology, College of Applied Medical Sciences, Taibah University, Madinah 41411, Saudi Arabia; 13Department of Internal Medicine, College of Medicine, Imam Abdulrahman Bin Faisal University, Dammam 34212, Saudi Arabia; 14Basic Medical Sciences, Majmaah University, Majmaah 11952, Saudi Arabia; 15Department of Infectious Disease, King Abdullah Medical City, Makkah 43442, Saudi Arabia; 16Department of Community Medicine and Health Care for Pilgrims, Faculty of Medicine, Umm Al-Qura University, Makkah 21955, Saudi Arabia; 17Department of Microbiology, Faculty of Life Sciences, University of Central Punjab, Lahore 54000, Pakistan; 18Department of Medical Microbiology and Parasitology, School of Medical Sciences, Universiti Sains Malaysia, Kubang Kerian 16150, Malaysia

**Keywords:** carbapenems, ESBL, carbapenem resistant *Enterobacteriaceae*, CRE, PCR, AST, antimicrobial resistance, AMR

## Abstract

*Enterobacteriaceae* have been classified as severely drug resistant bacteria by the World Health Organization due to their extensive production and dissemination of carbapenemases (CPs) and extended-spectrum β-lactamases (ESBL). The current study was conducted with the aim to determine the prevalence of CP- and ESBL-producing *Enterobacteriaceae,* as well as their antibiotic susceptibility profiles. For this, a hospital-based study was conducted which included 384 participants with bacterial infections. The collection and processing of specimens was conducted per standard microbiological protocol. The samples were inoculated on agar media plates to obtain the bacterial growths, and if they were positive for any bacterial growth, the antibiotic susceptibility testing was performed using disk diffusion method to check their antibiotic susceptibility patterns. The double disc diffusion as well as carbapenem inhibition techniques were used to examine the CP enzymes. Multiplex real-time PCR technique was performed to identify three distinct genetic types of CPs that have been identified in the *Enterobacteriaceae* (*KPC*, *NDM*, and *OXA-48*). A majority of participants (58.3%) in the current study were living in urban areas. A total of 227 (59.1%) patients were hospitalized. Furthermore, 26.04% of the patients were determined to be suffering from infections with *Enterobacteriaceae*. *Escherichia coli* was the most prevalent (9.1%) isolate overall, followed by *Klebsiella pneumoniae* (8.07%), *Acinetobacter baumannii* (2.6%), *Pseudomonas aeruginosa* (3.1%), *Enterobacter cloacae* (1.3%), *Proteus* spp. (1.3%), and *Morganella* spp. (0.5%). The studied patients were suffering from urinary tract infections (48.6%), blood stream infections (32.2%), wounds infection (11.9%), and respiratory infections (7.03%), confirmed with bacterial cultures. The resistance against carbapenems was seen in 31.4% of *E. coli* isolates, 25.8% in *K. pneumoniae*, 50% in *P. aeruginosa*, 25% in *A. baumannii*, and 20% in *E. cloacae* isolates. Such high rates of CP- and ESBL-producing *Enterobacteriaceae* are alarming, suggesting high spread in the study area. It is advised to implement better infection prevention and control strategies and conduct further nationwide screening of the carriers of these pathogens. This might help in reducing the burden of highly resistant bugs.

## 1. Introduction

Globally, infections caused by *Enterobacteriaceae* have a significant role in both mortality and morbidity [1]. The administration of antibiotics as part of empirical therapy is considered both the cause of increasing drug resistance and the solution of infections [2]. However, the development of drug resistance to such medicines has evolved into a major health issue globally [3]. The Center for Disease Control and Prevention (CDC), United States of America (USA), believes infections caused by carbapenem-resistant *Enterobacteriaceae* (CRE) to be an urgent concern to public health because they are being reported more often across the world [4]. The large percentage of *Enterobacteriaceae* is due to their being naturally occurring inhabitants of a human gastrointestinal (GI) tract, and the illnesses they cause result from being forced out of that environment. They can cause a variety of diseases, such as wound infections, septicemia, and lower respiratory tract infections (LRTIs) [5].

The extended-spectrum β-lactamases (ESBLs) have been seen as a serious threat to public health since the beginning of the century [6]. Carbapenems antibiotics belong to the β-lactam family. ESBLs can hydrolyze the majority of antibiotics belonging to the β-lactam family, except for carbapenems [7]. A recent study discovered that Pakistan has an unacceptably high incidence of ESBLs (over 80%) [8]. Carbapenems are one of the few treatments still available for *Enterobacteriaceae* that produce ESBLs [9]. However, bacterial infections that produce carbapenemase (CP) and are resistant to carbapenems have been chosen as a result of the excessive use of carbapenems to treat ESBL-positive pathogens [2].

Carbapenem antibiotics are used as a last line of defense against multi-drug resistant (MDR) bacteria [10,11]. The carbapenems are usually considered as last-line drugs, especially for the treatment of critically ill patients or those having a Gram-negative infection which is resistant to the majority of antibiotics [11]. In hospitals throughout Southeast and South Asia, Gram-negative bacteria have high rates for carbapenem resistance [7,9]. In this case, the polymyxins may be chosen for treatment, but there are also some reports which showed polymyxin resistance by a mcr-1 gene, making the problem worsened [12,13]. The rapid rise in the prevalence of resistance to carbapenem among the *Enterobacteriaceae* is connected with health care, particularly *Klebsiella pneumoniae (K. pneumoniae)* and *Escherichia coli* (*E. coli)* which are significant causes for infection and the antibiotic resistance burden [4,14].

The threat of CRE is mainly due to the emergence and spread of carbapenemase (CPs)-producing bacteria [15,16]. The majority of strains that develop carbapenem resistance have been reported to produce CPs [17,18,19]. A recent study from Punjab, Pakistan, has reported 14.4% CP-producing *Enterobacteriaceae* [20]. The increased consumption of antibiotics in clinical settings could also increase the selection pressure of antibiotic resistance genes and may lead to high antimicrobial resistance rates [6,21]. However, the growth of carbapenem resistance could also be caused by the consumption of fishery products [22]. There have been several reports of bacteria recovered from fisheries’ products with plasmids carrying genes for carbapenem resistance [23,24,25]. Commercial shrimp were found to include *Vibrio alginolyticus* carrying the encoding *VIM-1* and the encoding *NDM-1* genes in a Chinese retail market [22]. In Vietnam, *Enterobacteriaceae* carrying the *VIM-1* and *NDM-1,4,5, KPC* and *OXA-48* have been isolated from fish and other sea animals [25]. In Pakistan, the antimicrobial resistance rates are continuously increasing and leading to significant health threats [2,8]. The current study was conducted with the aims to see the prevalence of CP-, CRE-, and ESBL-producing *Enterobacteriaceae* both phenotypically and genotypically.

## 2. Results

### 2.1. Demographic Charateristics

During the study period, a total of 1206 patients were tested for possible bacterial infections from which 384 patients were found positive for various bacterial infections. Hence, the current study included a total of 384 individuals with bacterial infections. A majority of the patients were male (*n* = 210, 54.7%) while the remaining 45.2% were female. Participants in the current study varied in age group from 16 to 80 years. In addition, the majority of participants (58.4%) lived in urban areas. A total of 227 (59.1%) patients were hospitalized (Table 1). A total of 100 (26%) patients were found suffering from infections with *Enterobacteriaceae*. Urban residents had a much greater infection rate than those who lived in rural areas (Table 1).

The prevalence of isolated bacterial isolates is shown in Table 2*. E. coli* constituted the most prevalent isolate overall, following *K. pneumoniae*, *Acinetobacter baumannii* (*A. baumannii*), *Pseudomonas aeruginosa* (*P. aeruginosa*), *Enterobacter cloacae*, *Proteus* spp., and then *Morganella* spp. (Table 2). UTI, blood stream infections, wounds infection, and respiratory infection proportions with culture confirmation were 44%, 23%, 22%, and 11%, respectively.

### 2.2. Antibiotic Resistance Profiles

Amoxicillin-clavulanic acid, cefotaxime, was the antibiotic to which the majority of the *Enterobacteriaceae* showed the strongest resistance. Amoxicillin-clavulanic acid resistance among the isolates was found to be quite high, as high as 97.1%, in *E. coli*., and 100 % in *K. pneumoniae.* The majority of isolates showed tolerance to cefotaxime in terms of 3rd Generation cephalosporin (Table 3).

The antibiotic susceptibility patterns of ESBL-producing *Enterobacteriaceae* has been showed in Table 4. All of the ESBL-producing bacteria were completely resistant (100%) to Amoxicillin-clavulanic acid, cefotaxime, and ceftazidime. The *P. aeruginosa* isolates showed 100% resistance to carbapenems, piperacillin-tazobactam, and cefepime.

### 2.3. ESBL and Carbapenemase Production Profiles

Table 5 shows that the majority of isolates of *Enterobacteriaceae* were ESBL-producing. *Pseudomonas* spp. and *K. pneumoniae* produced more ESBL. The detected CP producers included *E. coli*, *K. pneumoniae*, and *E. cloacae*. In this study, a total of 28 carbapenem-resistant, Gram-negative isolates were evaluated. Of these, 24 of them, clearly described by PCR, contain the mentioned CP genes.

All of the CR *K. pneumoniae* isolates were found to generate CP. Blood had a greater level of carbapenem resistance compared to the other samples. In contrast to other samples, blood had a greater frequency of ESBL-producing isolates (Table 5).

### 2.4. Real-Time Multiplex PCR Analysis

A total of 28 *E. coli*, 29 *K. pneumoniae*, 5 *P. aeruginosa*, 5 *A. baumannii*, and 4 *E. cloacae* isolates were tested for *CTX*, *KPC*, *OXA-48*, and *NDM* genes. The prevalence of these genes in each tested bacterium is shown in Table 6.

## 3. Discussion

Over the years, an increased rate of ESBL producers have been reported among infected patients in hospital settings [26,27,28]. In this scenario, the therapeutic and clinical consequences of the introduction of ESBL-producing as well as CR *Enterobacteriaceae* cause a drastic reduction in the treatment options [29]. Only limited options to treat these Gram-negative bacterial infections are available as they are widely resistant to most common antibiotics. The current study was conducted with the aims to see the prevalence of *Enterobacteriaceae* in terms of CP and ESBL production and their antibiotic susceptibility patterns in an underdeveloped area of Punjab, Pakistan.

From a total of 1206 patients, 384 were found positive for different bacterial infections. From these 384, a total of 100 isolates were belonging to *Enterobacteriaceae.* Most of the participants of the current study who were infected with *Enterobacteriaceae* were living in urban areas (65%) as compared to rural areas (35%). This could be connected to changes in the way patients are exposed to antibiotics, which is a key contributor to the development of antibiotic resistance. A similar previous study from Japan used multi drug resistant *E. coli* as an indicator for establishing antibiotic resistance and its direct impact on public health [30].

In the current study, compared to outpatients, inpatients had a considerably greater proportion (59.1%) of infections. In the current study, 23% of the *Enterobacteriaceae* were CP-producers. In a systematic assessment from Africa, eight distinct studies found a high prevalence of *K. pneumoniae* and *E. coli* that produce CP [31]. The prevalence of *Enterobacteriaceae* that produce ESBL was 67% in the current study which is consistent with an Iranian study (40.8%) [32] and an Indian study (48.2%) [33]. However, in Ethiopia, the prevalence of ESBL was 28.2% [34]. *K. pneumoniae* were among the most common producers of ESBLs in our investigation, even greater than that of the Ethiopian study (54.5%) [35] and for Burkina Faso (58%) [36]. The level of ESBL-producing *E. coli* in this research is comparable to a study conducted in Benin (36.2%) [37].

It is known that *Enterobacteriaceae* may acquire resistant plasmid from other bacterial species as well. Hence, the presence of a resistance gene poses a greater risk as it can be transmitted to potentially susceptible organisms. In Uganda, there is a lower prevalence of ESBL producers (28.1%) [38]. According to previous studies, there may be a larger colonization of *Enterobacteriaceae* in health facilities, which then, in turn, promotes the proliferation of ESBL- and CP-producing genes in strains linked to the medical industry [39]. Even if the severity of ESBL and CRE varies, all data revealed an increase in ESBL isolates in emerging economies, which may be related to the extensive utilization of cephalosporins, lax antibiotic use regulation, and empirical treatment. On the other hand, the administration of carbapenems in hospitalized patients, especially in surgical ICU, medical ICU, and other medical wards, is very common, and this might be a contributing factor for increasing carbapenem resistance in such cases [40].

In the current study, the *E. coli* isolates have shown a high resistance to amoxicillin-clavulanic acid (97.1%), cefotaxime (94.2%), and ceftazidime (91.4%). Contrarily, a study from Ethiopia [41] also showed higher resistance to amoxicillin-clavulanic acid (90.3%), amoxicillin-clavulanic acid (85.7%), and cefotaxime (56.5%). Another study has reported equal rates of ceftazidime (33%) and cefotaxime (31%) resistance [42]. The carbapenem resistance rates in the current study ranged from 20–30% among *Enterobacteriaceae*, which is relatively lower than that reported in Sindh province of Pakistan, around 59% [43]. Another study from Lahore, Punjab, reported a higher rate of carbapenem resistance in *A. baumannii* (85.8%) [5].

The results of the current study have shown that there was a difference in the antibiotic susceptibility patterns of ESBL- and non-ESBL-producing organisms. The overall carbapenem resistance was seen in 31.4% of non-ESBL-producing *E. coli*, while in the case of ESBL-producing *E. coli*, the carbapenem resistance rate was 39.2%. The non-ESBL-producing *K. pneumoniae* isolates have shown 25.8% carbapenem resistance while 27.5% resistance to carbapenems was seen in ESBL-producing *K. pneumoniae* isolates. A similar previous study from Pakistan has reported that the presence of ESBL-producing *E. coli* was 25.5%, from which around 20% of the isolates were resistant to carbapenems [44]. Another study from Rawalpindi, Pakistan, showed that an overall 19.3% were carbapenem-resistant [45].

A previous study from Rawalpindi, Pakistan, has reported that most of the CP-producing organisms were *E. coli* (86%) [20]. Another study has reported that CP genes were detected in 61 out of the 72 carbapenem-resistant isolates [45]. Results of the current study has showed that the most CP-producing organism was *P. aeruginosa* (30%)*,* followed by *K. pneumoniae* (25.8%), *E. coli* (25.7%), *A. baumannii* (16.6%), and *E. cloacae* (20%). Furthermore, it was found in the current study that the prevalence of all drug-resistant bacteria was higher in urban areas as compared to the rural areas.

It was found in the current study that most of the *Enterobacteriaceae* were positive for the *NDM* gene followed by *CTX, KPC*, and *OXA-48*. Similar results were seen in another study from Pakistan, in which the most common CP-encoding gene was *NDM* [20]. Another study from Pakistan reported that the *CTX* gene was detected in 74.8% of the *E. coli* isolates [44]. The prevalence of CP and ESBL-producing *Enterobacteriaceae* are alarmingly increasing in the study area. Thus, improving the infection prevention strategy and further large-scale studies at the national level are recommended in order to prevent another pandemic.

### Study Limitations

Because of ethical approval restriction, the data related to the total hospital days of patients, days of antibiotic therapy, and previous history of the antibiotic usage could not be added in the current study.

## 4. Materials and Methods

### 4.1. Ethical Approval

Before starting sample and data collection, signed informed consents were obtained from study participants. Study participants were informed about how the study will be conducted and their potential benefits for the community. Participants who did not give full information and/or permission to use their data were eliminated from the study. An ethical approval from the Institutional Research Board was obtained.

### 4.2. Study Plan and Timeframe

A regional research study was conducted from February 2022 to April 2022, in a secondary care hospital of district Vehari, Punjab, Pakistan. It has more than 1000 medical and non-medical staff members, and it includes 430 beds across medical, surgical, orthopedic, and pediatric wards. An average of 600 patients receive medical care each day.

### 4.3. Sample Size and Sampling Method

Applying v 3.5.1 of Epi info (publicly available program, http://www.cdc.gov, Accessed on 14 January 2022), the size of the sample was calculated by taking into account a 95% level of confidence, marginal variance (5%), as well as a ratio of 0.5% ESBL. Therefore, a total sampling size of 384 was determined. Until the needed sample size had been reached, patients who had been clinically supposed of having various infections were recruited. Participants in the research who could not provide full data, had an inadequate sample (Sputum/saliva), or provided an inadequate quantity of all samples were excluded.

### 4.4. Data Collection

One-on-one interviews were conducted with a questionnaire to collect the information on demographic factors along with individuals’ health records. The data related to the residence area of the patients were also collected in order to check whether there is any difference in terms of AMR rates. The residential area was categorized into rural and urban areas. The rural areas were the villages, towns, and small cities which have a few thousand inhabitants and were mostly lacking in amenities of civic and social life including fewer healthcare services while the urban areas were the well-developed cities with comparatively more population and more facilities, especially in terms of healthcare services.

### 4.5. Collection and Processing of Specimens

All of the samples from study participants were collected by following the standard microbiological techniques to minimize the risk of contamination. The urine samples were collected in tight capped, wide-mouth, sterile containers, while the blood samples (10 mL for adult, 5 mL for infants, and 2 mL for newborns) were collected in BACT/ALERT^®^ culture media, aerobic and anaerobic vials (Biomerieux, Marcy-l’Étoile, France). For the collection of sputum sample, the participants were asked to wash their mouths using water, and then 2 mL of purulent sputum, inside a sterile container, was collected. Watery saliva Sputum samples were not processed utilizing microbiological techniques. Samples from the wound were collected with a syringe or a sterile swab, while the pus, purulent exudates, and discharge were aseptically taken out from base of the lesion. Brain–heart infusion transport medium was used to submerge the swab. After the collection of samples, these were packed in the sterile zipper plastic bags and kept in the tight capped container containing ice bags (2–8 °C) to transport into the microbiology section of the pathology laboratory for further processing.

After receiving the samples in the microbiology section, the containers were opened, and samples were taken out for further processing in the biosafety cabinet. The urine samples were inoculated on cysteine-lactose-electrolyte-deficient (CLED) agar media (Oxoid Limited, Hampshire, UK) with the help of a sterile disposable wire loop. The aseptically collected venous blood samples were placed in the Bact/Alert automated blood culture system (Biomerieux, Marcy-l’Étoile, France). After the instrument gave a notification for a positive blood culture, the samples were taken out from the instrument and reinoculated on blood, chocolate, and MacConkey (MAC) agar plates (Oxoid Limited, Hampshire, UK). The sputum, pus, wound swabs, and other samples were inoculated on blood, chocolate, and MAC agar plates.

After the inoculation of samples on various agar media plates, these were incubated at 37 °C for 18–24 h. The inoculated blood, CLED, and MAC plates were incubated under aerobic conditions, while the chocolate agar plates were incubated under anaerobic or microaerophilic conditions to obtain the growth of fastidious organisms. If there was no bacterial growth obtained after the first incubation period, the negative plates were incubated again for over 24 h. The inoculated plates were reported negative for bacterial growth if no growth had been obtained in 48 h of incubation period.

### 4.6. Identifying Bacterial Isolates

Using standard microbiology laboratory techniques, all isolates of the family *Enterobacteriaceae* were identified. Final identification of bacterial isolates was conducted using the Gram staining and biochemical tests. The biochemical tests used to identify the isolates were indole, the triple sugar iron (TSI), citrate, urease, oxidase, and motility tests [46].

### 4.7. Antimicrobial Susceptibility Testing

The clinical and laboratory standards institute (CLSI) recommended Kirby–Bauer disc diffusion method was applied to determine the susceptibility of all detected isolated bacteria to various antimicrobial agents on the Mueller–Hinton agar (MHA) plates. To obtain bacterial inoculum comparable to the McFarland turbidity (0.5) standard [47], pure bacterial colonies were suspended inside a tube of normal saline using 4 mL. The suspended colonies were uniformly spread on a Mueller–Hinton agar using a sterile swab before putting the discs on the MHA plate [47].

The mentioned antibiotic discs were used to test each isolate: amoxicillin (10 μg), amoxicillin-clavulanic acid (20 μg), cefotaxime (30 μg), ceftazidime (30 μg), cefepime (30 μg), meropenem (10 μg), imipenem (10 μg), gentamicin (10 μg), Amikacin (10 μg), Nitrofurantoin (10 μg), Piperacillin-tazobactam (10 μg), and sulfamethaxazole-trimethoprin (30 μg). The petri plates were incubated for 24 h at 37 °C. A digital caliper was used to measure the zone of inhibitions (ZOIs) surrounding the discs [48].

Depending on the standardized chart provided by CLSI recommendations [47], the antibiotics susceptibility tests were classified as sensitive, moderate, or resistant. If the isolate proved resistant to a minimum of one antibiotic in three different antimicrobial groups, the presence of multi-drug resistant *Enterobacteriaceae* was investigated.

### 4.8. Screening of ESBLs by Phenotypic Method

At first, the testing for ESBLs was performed by determining the diameters of the ZOIs caused either by amoxicillin-clavunate, cefotaxime, cefepime, or ceftazidime (30 μg) on the MHA plates [47]. According to the recommendation of CLSI, the ZOI breakpoints for amoxicillin-clavunate, ceftazidime, cefepime, and cefotaxime were 22 mm, 25 mm, and 27 mm, respectively [48].

### 4.9. Detection of Carbapenemase Producers

The modified carbapenem inhibitory method (mCIM), advised by CLSI [47], was used to evaluate bacterial isolates that had complete or intermediate resistance to at least one of the carbapenem (meropenem 10 μg, imipenem 10 μg) in the above-mentioned disc diffusion test. Therefore, the imipenem disc (10 μg) was administered after homogenizing the isolates in 3 mL of tryptic soya broth (TSB); the mixture was then cultured at 37 °C in the presence of climate air over 4 h. Meropenem and imipenem in TSB were then administered after the McFarland standards equivalent solution of the indicator organism (*E. coli* ATCC 25922), which is carbapenem sensitive, then swabbed onto a Mueller–Hinton agar. Zones of inhibition for carbapenems and other drugs were assessed after a 24-h incubation period at 37 °C. It was determined that a bacterial strain was a carbapenemases generator if it had pinpoint colonies inside an inhibition zone of 6 to 15 mm or even an inhibition zone of 16 to 18 mm and did not inhibit the *E. coli* ATCC 25922, which is carbapenem-susceptible [49].

As a quality control, throughout data collection, the data were examined to ensure that they were accurate and fully recorded on the spreadsheet. All laboratory tests were performed under strict adherence to quality assurance guidelines. While incubating 5% from the batch for an overnight period at 35–37 °C, media sterility was examined. Before usage, the medium, reagents, as well as antibiotic discs were examined for their expiration dates. Throughout the investigation, referent strains of *Staphylococcus aureus* (ATCC 25923), *E. coli* (ATCC 25922), *Enterococcus faecalis* (ATCC 29212), and *P. aeruginosa* (ATCC 27853) were utilized to evaluate the media capacity produced and to sustain growth of bacteria for cultures and susceptibility tests. *E. coli* ATCC 25922 and *K. pneumoniae* ATCC 700603 were chosen as negative and positive controls, accordingly, for the ESBL detection process. As negative and positive quality assurance isolates for detection of CPs, *K. pneumoniae* ATCC BAA1705 as well as *K. pneumoniae* ATCC BAA1706 were selected, respectively. The CLSI standards [47] were used to interpret the findings.

### 4.10. Gene Confirmation through Real Time PCR

Considering the manufacturer’s advice for quick identification and discrimination of the *CTX*, *blaNDM*, *blaOXA-48*, and *blaKPC* genes’ sequences associated with carbapenem resistance amongst gram-negative bacteria, all verified strains of CRE from the culture were examined using the Conventional Polymerase Chain Reaction [50].

#### 4.10.1. Extraction of DNA

Isolation of the bacterial DNA was conducted by utilizing the NucliSens easyMAG platform and NucliSens magnetic extraction reagents (bioMérieux, Craponne, France). Extraction buffer (60 L) was used to recover the isolated DNA [1]. The extracted DNA was shifted to sterilized Eppendorf tube, and eluted DNA was, finally, stored at 4 °C for storage and further processing. After the extraction of DNA, the agarose gel electrophoresis was performed to check the presence of DNA. 1% agarose gel was prepared in 50X Tris-acetate-EDTA (TAE) buffer.

#### 4.10.2. Primer Designing

The information about the baseline genes utilized in this test was found on the website http://www.lahey.org/studies/ (Accessed on 16 January 2022). Carbapenemases (Class A), which code for KPC type; oxacillinases (Class D), which code for OXA-48; and metalloenzymes (Class B), which code for NDM, IMP, and VIM, were the genes in question [10]. The GenBank (http://www.ncbi.nlm.nih.gov/genbank/) (Accessed on 16 January 2022) was used to retrieve these gene sequences of *blaCTX*, *blaKPC*, *blaOXA-48*, *blaNDM-1*. Primers were created particularly to amplify every allele of each carbapenemase family of genes listed above based on the thorough studies as well as alignments of every carbapenemase type. The Lasergen software programmer (DNASTAR, Madison, WI) was used to figure out the melting point (Tm) of each gene from the carbapenemase family that was amplified. Primers *CTX*, *IMP*, *KPC*, and *NDM* pairs then were modified from formerly released sequences [31]. The primer was tested in a single Reaction PCR configuration. In order to verify that the primers were successfully amplified at their particular locus, its amplicon displayed the predicted Tm in order to verify the accuracy of the real-time PCR. Structure of the Multiplex was then improved by measuring various primer pair concentrations. A 2% agarose gel in electrophoresis was used to confirm the length of every PCR result. IDT created all of the primers (Coralville, IA, USA). Table 7 lists primer sequences with references.

#### 4.10.3. Multiplex Real Time PCR

The amplifications of desired genes were conducted using real-time polymerase chain reactions conducted in 25 mL of Master Mix containing distilled water, primer, and 1 mL of DNA template. The PCR conditions were taken from the previously published studies [51,52].

Each of the positive controls were evaluated in quadruplicate within the same cycle and in three other runs in order to evaluate the data from this novel multiplex, real-time PCR method. In order to determine how consistently the assay performed across and within runs, means of the melting temperature, standard deviation, and variation coefficient were computed [53].

### 4.11. Statistical Analysis

At first, the data were entered into a Microsoft Excel sheet (Microsoft, Redmond, WA, USA) and then transferred to a Statistical Package for Social Science (SPSS) version 22.0 (IBM USA) spreadsheet. Frequencies (*n*) and percentages (%) were calculated and summarized in tables to show the frequency of demographic characteristics. The Fisher’s exact test was run to see the statistical differences among different variables. A *p*-value of <0.05 was considered statistically significant.

## 5. Conclusions

Results of the current study have shown an alarming rise in antibiotic resistance among *Enterobacteriaceae* along with ESBL- and CP-producing characteristics which might lead to a higher prevalence of MDR bacteria in near future. Another major issue reported in the current study was resistance to 3rd and 4th generation cephalosporins. Therefore, actions to improve the infection prevention and control measures must be taken. Policy makers of the country must take increasing antibiotic resistance into account and issue guidelines for better management of such patients infected with MDR, XDR and CREs. There should be rules for selling, purchasing and prescribing the antibiotics in order to minimize the usage of antibiotics. The antibiotics should only be prescribed after seeing the bacterial culture reports and their antibiotic susceptibility patterns. Furthermore, large-scale national surveillance on the profile of carbapenem resistance, CP production, and ESBL production and their determining genes among *Enterobacteriaceae* clinical isolates is required to adjust the routine use of antimicrobials.

## Figures and Tables

**Table 1 antibiotics-12-00148-t001:** Demographic characteristics and infections with *Enterobacteriaceae* that were found positive in a culture using various clinical samples.

Variable	Total Samples (*n*)	Positive Samples for *Enterobacteriaceae* (*n*)	Prevalence (%)	*p*-Value
Age (years)				0.63
<18	59	11	2.8	
18–28	63	14	3.6	
29–38	58	14	3.6	
39–48	65	14	3.6	
49–60	45	13	3.3	
>60	94	34	8.8	
Gender				0.35
Female	174	48	12.4	
Male	210	52	13.5	
Area of residence				<0.05 *
Urban	160	65	16.9	
Rural	224	35	9.1	
Admission department				<0.05 *
ICU	92	27	7.0	
Medical	107	43	11.1	
Surgery	28	13	3.3	
OPD	157	17	4.4	
Hospital setting				<0.01 *
In–patient	227	83	21.6	
Out–patient	157	17	4.4	
**Total**	384	100	26.0	

ICU: Intensive care unit. OPD: outpatient department. * Statistically significant. *n*: numbers. %: percentage.

**Table 2 antibiotics-12-00148-t002:** Prevalence of *Enterobacteriaceae* in clinical specimens taken from various infection sites, hospitalization statuses, and residence areas of patients.

Variables	*E. coli*	*K. pneumoniae*	*P. aeruginosa*	*A. baumannii*	*E. cloacae*	*Proteus* Spp.	*Morganella* Spp.
Specimen type							
Urine (*n* = 187)	22 (11.7%)	7 (3.7%)	7 (3.7%)	4 (2.1%)	1 (0.5%)	2 (1.0%)	1 (0.5%)
Blood (*n* = 124)	3 (2.4%)	13 (10.4%)	2 (1.6%)	3 (2.4%)	2 (1.6%)	-	-
Sputum (*n* = 27)	2 (7.4%)	5 (18.5%)	-	2 (7.4%)	1 (3.7%)	1 (3.7%)	-
Wound Sample (*n* = 46)	8 (17.3%)	6 (13.04%)	1 (2.1%)	3 (6.5%)	1 (2.1%)	2 (4.3%)	1 (2.1%)
Hospitalization status							
ICU (*n* = 92)	10 (10.8%)	5 (5.4%)	3 (3.2%)	6 (6.5%)	1 (1.0%)	1 (1.0%)	1 (1.0%)
Medical (*n* = 107)	14 (13.0%)	16 (14.9%)	4 (3.7%)	2 (1.8%)	3 (2.8%)	3 (2.8%)	1 (0.9%)
Surgery (*n* = 28)	6 (21.4%)	4 (14.2%)	-	3 (10.7%)	-	-	-
OPD (*n* = 157)	5 (3.1%)	6 (3.8%)	3 (1.9%)	1 (0.6%)	1 (0.6%)	1 (0.6%)	-
Area of residence							
Urban (*n* = 160)	19 (11.8%)	14 (8.7%)	6 (3.7%)	5 (3.1%)	3 (1.8%)	3 (1.8%)	-
Rural (*n* = 224)	16 (7.1%)	17 (7.5%)	4 (1.7%)	7 (3.1%)	2 (0.8%)	2 (0.8%)	2 (0.8%)
**Total** (*n* = 384)	35 (9.1%)	31 (8.0%)	10 (2.6%)	12 (3.1%)	5 (1.3%)	5 (1.3%)	2 (0.5%)

ICU: Intensive care unit. OPD: outpatient department. *n*: numbers. %: percentage.

**Table 3 antibiotics-12-00148-t003:** Antimicrobial resistance profiling of *Enterobacteriaceae*.

Antimicrobials	*E. coli* (*n* = 35) (*n*/%)	*K. pneumoniae* (*n* = 31) (*n*/%)	*P. aeruginosa* (*n* = 10) (*n*/%)	*A. baumannii* (*n* = 12) (*n*/%)	*E. cloacae* (*n* = 5) (*n*/%)
Amoxicillin–clavulanic acid	34 (97.1)	31 (100)	-	1 (8.3)	5 (100)
Amikacin	20 (57.1)	21 (67.7)	1 (10)	-	-
Nitrofurantoin	25 (71.4)	23 (74.1)	-	-	1 (20)
Sulfamethaxazole-trimethoprin	26 (74.2)	23 (74.1)	-	4 (33.3)	4 (80)
Gentamicin	21 (60.0)	19 (61.2)	4 (40)	3 (25)	1 (20)
Chloramphenicol	25 (71.4)	26 (83.8)	-	9 (75)	0
Cefotaxime	33 (94)	30 (96.7)	-	-	5 (100)
Ceftazidime	32 (91.4)	29 (93.5)	6 (60)	4 (33.3)	4 (80)
Cefepime	26 (74.2)	29 (93.5)	5 (50)	4 (33.3)	4 (80)
Meropenem	11 (31.4)	8 (25.8)	5 (50)	3 (25)	1 (20)
Imipenem	11 (31.4)	8 (25.8)	5 (50)	3 (25)	1 (20)
Piperacillin-tazobactam	22 (62.8)	24 (77.4)	6 (60)	3 (25)	1 (20)

*n*: numbers. %: percentage.

**Table 4 antibiotics-12-00148-t004:** Antimicrobial resistance profiling of ESBL-producing *Enterobacteriaceae*.

Antimicrobials	*E. coli* (*n* = 28) (*n*/%)	*K. pneumoniae* (*n* = 29) (*n*/%)	*P. aeruginosa* (*n* = 5) (*n*/%)	*A. baumannii* (*n* = 5) (*n*/%)	*E. cloacae* (*n* = 4) (*n*/%)
Amoxicillin–clavulanic acid	28 (100)	29 (100)	-	5 (100)	4 (100)
Amikacin	18 (64.2)	19 (65.5)	1 (20)	-	-
Nitrofurantoin	21 (75)	22 (75.8)	-	-	1 (25)
Sulfamethaxazole-trimethoprin	22 (78.5)	22 (75.8)	-	4 (80)	4 (100)
Gentamicin	17 (60.7)	16 (55.1)	4 (80)	3 (60)	1 (25)
Chloramphenicol	22 (78.5)	24 (82.7)	-	5 (100)	0
Cefotaxime	28 (100)	29 (100)	-	5 (100.)	4 (100)
Ceftazidime	28 (100)	29 (100)	-	5 (100)	4 (100)
Cefepime	21 (75)	22 (75.8)	5 (100)	4 (80)	4 (100)
Meropenem	11 (39.2)	8 (27.5)	5 (100)	3 (60)	1 (25)
Imipenem	11 (39.2)	8 (27.5)	5 (100)	3 (60)	1 (25)
Piperacillin-tazobactam	20 (71.4)	22 (75.8)	5 (100)	3 (60)	1 (25)

*n*: numbers. %: percentage.

**Table 5 antibiotics-12-00148-t005:** Prevalence of ESBL, carbapenem resistance, and carbapenemase producers.

Characteristics		ESBL Producers (*n*, %)	Carbapenems Resistance (*n*, %)	Carbapenemase Producers (*n*, %)
Hospitalization status				
*E. coli*	ICU	6, 17.1	5, 14.2	3, 8.5
	Medical	8, 22.8	1, 2.8	3, 8.5
	Surgery	3, 8.5	2, 5.7	1, 2.8
	OPD	11, 31.4	3, 8.5	2, 5.7
*K. pneumoniae*	ICU	8, 25.8	3, 9.6	2, 6.4
	Medical	6, 19.3	2, 6.4	3, 9.6
	Surgery	5, 16.1	2, 6.4	1, 3.2
	OPD	10, 32.2	1, 3.2	2, 6.4
*P. aeruginosa*	ICU	2, 20.0	2, 20.0	2, 20.0
	Medical	1, 10.0	-	-
	Surgery	1, 10.0	2, 20.0	-
	OPD	1, 10.0	1, 10.0	1, 10.0
*A. baumannii*	ICU	2, 16.6	1, 8.3	1, 8.3
	Medical	-	1, 8.3	-
	Surgery	1, 8.3	-	
	OPD	2, 16.6	1, 8.3	1, 8.3
*E. cloacae*	ICU	2, 40.0	-	-
	Medical	1, 20.0	-	-
	Surgery	-	-	-
	OPD	1, 20.0	1, 20.0	1, 20.0
Area of residence				
*E. coli*	Urban	17, 48.5	6, 17.1	6, 17.1
	Rural	11, 31.4	5, 14.2	3, 8.5
*K. pneumoniae*	Urban	16, 51.6	4, 12.9	5, 16.1
	Rural	13, 41.9	4, 12.9	9.6
*P. aeruginosa*	Urban	3, 30.0	2, 20.0	2, 20.0
	Rural	2, 20.0	3, 30.0	2, 20.0
*A. baumannii*	Urban	4, 33.3	1, 8.3	1, 8.3
	Rural	1, 8.3	2, 16.6	1, 8.3
*E. cloacae*	Urban	3, 60.0	1, 20.0	1, 20.0
	Rural	1, 20.0	-	-

ICU: Intensive care unit. OPD: outpatient department. *n*: numbers. %: percentage.

**Table 6 antibiotics-12-00148-t006:** Number of genes detected against each bacterium.

Bacteria	Genes	Detection (*n,* %)
*E. coli*	*KPC*	4, 14.2
	*NDM*	8, 28.5
	*CTX*	8, 28.5
	*OXA-48*	11, 39.2
*K. pneumoniae*	*KPC*	6, 20.6
	*NDM*	13, 44.8
	*CTX*	7, 24.1
	*OXA-48*	5, 17.2
*P. aeruginosa*	*KPC*	3, 60
	*NDM*	2, 40
	*CTX*	1, 20
	*OXA-48*	2, 40
*A. baumannii*	*KPC*	2, 40
	*NDM*	1, 20
	*CTX*	1, 20
	*OXA-48*	2, 40
*E. cloacae*	*KPC*	1, 25
	*NDM*	1, 25
	*CTX*	2, 50
	*OXA-48*	1, 25

**Table 7 antibiotics-12-00148-t007:** Detection with PCR of ESBL and carbapenemases genes.

Target Genes	Primer Name	Sequence (5′–3′)	Amplicon Size	Primer Concentration	Reference
*blaCTX*	CTX-M CTX-R	TGGGTAAAATAGGTGACCAGA ATGTGCAGCACCAGTAAGGT	650	0.1	[51]
*blaKPC*	KPC-F KPC-R	CGTCTAGTTCTGCTGTCTTG CTTGTCATCCTTGTTAGGCG	798	0.5	[52]
*blaNDM-1*	NDM-F NDM-R	GGTTTGGCGATCTGGTTTTC CGGAATGGCTCATCACGATC	621	0.1	[52]
*blaOXA-48*	OXA-48-F OXA-48-R	GCGTGGTTAAGGATGAACAC CATCAAGTTCAACCCAACCG	438	0.2	[52]

## Data Availability

More data related to the current study is available upon a reasonable request to namalik288@gmail.com.

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
