# Peer review of "Prevalence of Carbapenemase and Extended-Spectrum β-Lactamase Producing Enterobacteriaceae: A Cross-Sectional Study"

_antibiotics, 2023, doi:10.3390/antibiotics12010148_

Round 1

Reviewer 1 Report

Summary

The submitted paper presents the prevalence of Carbapenamese and ESBL producing gram-negative bacteria in a hospital cohort. The authors identify the prevalence of various carbapenem resistant bacteria and their susceptibility to various antibiotics.

Comments:

Line 91: CP is not defined prior to using the acronym.

In the Results section when describing the cohort please include the location and the definitions of Urban and Rural, since these definitions are country dependent in terms of healthcare access.

In Table 1 and Table 2, when showing group comparisons, it is generally recommended that a test statistic or p-value is displayed. I would recommend the authors try Fisher's exact test to show group comparisons.

In Table 3 Carbapenems resistance and Carbapenems Producers follow the same definition and have the same values. This seems a bit redundant.

Table 5 needs to display the number of samples from each species. The percentages alone are hard to understand.

In the abstract the authors claim:

"To achieve the best possible usage of antibiotics, it is advised to implement better infection prevention and control strategies and conduct further nationwide screening of the carriers of these pathogens. "

There is no information on antibiotic usage for the given cohort, so how does the current paper help improve prevention and control strategies?

Suggestions:

This paper can be an important addition to the literature surrounding Carbapenem resistance in developing countries. The authors do a good job of contextualizing their findings in relation to other studies performed in Iran, India and several African nations.

The statistical analysis of the data needs to be improved. Publications reporting prevalence in hospital cohorts tend to use patient days, or days of antibiotic therapy, as a normalizing factor to display rates.

Using raw counts and percentages not wrong, but the authors need to perform at least some preliminary statistical tests to show significance of results.

My recommendation for the authors would be to incorporate a statistician in the study to help elaborate their results.

Author Response

Reviewer 1

Comments and Suggestions for Authors

Summary

The submitted paper presents the prevalence of Carbapenamese and ESBL producing gram-negative bacteria in a hospital cohort. The authors identify the prevalence of various carbapenem resistant bacteria and their susceptibility to various antibiotics.

Comments:

Line 91: CP is not defined prior to using the acronym.

Response: Line 94: The abbreviation for CPs has been defined.

In the Results section when describing the cohort please include the location and the definitions of Urban and Rural, since these definitions are country dependent in terms of healthcare access.

Response: Line 258-264: Dear reviewer, thank you for your valuable suggestion. The description has been added in the material and methods section.

In Table 1 and Table 2, when showing group comparisons, it is generally recommended that a test statistic or p-value is displayed. I would recommend the authors try Fisher's exact test to show group comparisons.

Response: Dear reviewer, thank you for your valuable suggestion. We have run the statistical analysis on table 1 in the revised version of manuscript. In table 2, we couldn’t run because of some missing values.

In Table 3 Carbapenems resistance and Carbapenems Producers follow the same definition and have the same values. This seems a bit redundant.

Response: Dear reviewer, we really appreciate that you have noted our mistake. It was mistakenly written as same. The correct values have been mentioned in table 3 of revised manuscript.

Table 5 needs to display the number of samples from each species. The percentages alone are hard to understand.

Response: The values has been added in table 6.

In the abstract the authors claim:

"To achieve the best possible usage of antibiotics, it is advised to implement better infection prevention and control strategies and conduct further nationwide screening of the carriers of these pathogens. "

There is no information on antibiotic usage for the given cohort, so how does the current paper help improve prevention and control strategies?

Response: Line 56: Dear reviewer, we appreciate your that you have noted a technical point. We have revised the conclusive remarks in the revised version of manuscript.

Suggestions:

This paper can be an important addition to the literature surrounding Carbapenem resistance in developing countries. The authors do a good job of contextualizing their findings in relation to other studies performed in Iran, India and several African nations. The statistical analysis of the data needs to be improved. Publications reporting prevalence in hospital cohorts tend to use patient days, or days of antibiotic therapy, as a normalizing factor to display rates. Using raw counts and percentages not wrong, but the authors need to perform at least some preliminary statistical tests to show significance of results. My recommendation for the authors would be to incorporate a statistician in the study to help elaborate their results.

Response: Dear reviewer, we would like to appreciate for your valuable suggestion. We have added the statistical analysis part in the revised version of manuscript at line 394-400. However, your suggestion to add the patient’s hospitalization data or days of antibiotics data, we are unable to add because of certain limitations in the ethical approval from the parent institution. Hence, we have added this as one of the study limitations (Line 233-235).

Reviewer 2 Report

Abstract:

·         Extended-spectrum β-lactamases (ESBL).

·         Line 48: urinary tract infection (UTI)

·         Please explain the methodology in abstract shortly and clearly.

·         Please amend the conclusion according to the result.

Introduction:

·         This study is a fantastic one; thus, more papers should be used as strong references for this manuscript. Due to this fact, I recommend the authors to read and add the following papers to the References section of the manuscript to have fruitful Introduction and Discussion sections:

doi.org/10.1155/2022/5727638, doi.org/10.3390/pathogens11091019, doi.org/10.1128/spectrum.02137-22, doi: 10.2147/IDR.S356489, 10.1007/978-3-030-76320-6_20

·         The introduction is too general: Report the epidemiology of carbapenemases and ESBLs that focus on the study area. Please show the study gaps.

·         Line 80: I would like to write β-lactamases, not b-lactamases. Please amend this.

·         Strong statements have been given in “Line 80-81” of the introduction with no references to support them. Please provide references

·         Firstly, please write down the name of all bacteria into italic. Secondly, use the full name of bacteria in first i.e. Staphylococcus aureus (S. aureus) and then you can use the short name of bacteria i.e. S. aureus.

·         Check reference number 7: Use an appropriate reference for nosocomial infection.

Methodology

·         Line 201: Extended Spectrum β-lactamase (ESBLs)

·         Not clear regarding sample collection, transportation and preservation: How did you collect the blood sample? How did you transport the samples into microbiology lab? Did you preserve the sample? If yes, then for how many days and what was the temperature?

·         Section 4.5: Line 213, Cysteine Lactose Electrolyte Deficient agar?

Provide the manufacturer details of each chemical/material used (Company, City, Country).

·         Why did inoculate urine samples on Blood and MacConkey agar? Please also give reference.

·         “Positive samples were processed as per protocol and sample with no noticeable growth were ………………….. which it was declared negative” Not clear. Please modify this.

·         Line 239: Enterobacteriaceae? Please write in italic.

·         Line 241: indole production? TSI agar for Hydrogen Sulphide? citrate usage? urease test? oxidase? carbohydrate utilization assays?

·         Section 4.7: Clinical and Laboratory Standards Institute (CLSI)

·         In the method and result part, the authors used disk diffusion to do the antimicrobial activity. Is it about the measurement of zone inhibition or something else? This needs to be clarified.

·         Section 4.7: Please give the reference why 0.5 MacFarland was used in AST?

·         Section 4.7: Rephrase: The brackets are confusing, directly report antibiotic discs without classes or use table. Gentamicin (CN, 10μg)

·         Section 4.8: Screening of ESBLs by Phenotypic method: Please modify this heading. This section is not clear and creating confusion for reader. Please revise this accordingly what method was used for ESBLs identification?

·         Section 4.9: Please write down the name of bacteria in italic (2nd paragraph).

·         Section 4,10: Please write down the name of genes in italic (CTX, blaNDM, blaOXA-48, and blaKPC genes sequences).

·         Section 4.10.1: how did you confirm the DNA extraction either by spectrophotometer or gel electrophoresis?

·         Did you preserve the DNA sample? If yes, what was SOPs followed?

·         Section 4.10.3: Give references for the methods or provide details: What concentrations of dNTPs? Taq polymerase? Minerals? etc.   

·         Table 6: Please give the “Stands for” in legend. Tm? Please write the name of gene in italic.

Results:

·         Section 2.1: What was the total number of samples among which 384 were positive for bacterial infection? Please mention.

·         What about female? Please modify this accordingly.

·         Table 1: ICU? OPD? Please mention legend.

·         Section 2.2: what was the exact number of ESBL, CPR and CRE producing bacteria? Please mention it clearly and give along with % in Table 1.

·         Table 4: What about ESBL and CRE producing bacteria? Please also mention in two separate tables.

Discussion:

·         Revise the discussion accordingly. What did you expect?

·         Avoid much repetitions of the results, please follow the STROBE guidelines

·         Please use the recent studies conducted in Pakistan and neighbouring countries to make fruitful Discussion.

·         Conclusion is not coming from Results. Please revise this. What about less resistant antibiotic either belong to carbapenem group or non-carbapenem group antibiotic? Could physician suggest less resistant antibiotics to treat infection? What was the conclusion of this study that will be beneficiary for Physician researcher and population? Please explain it clearly in conclusion.

Author Response

Reviewer 2

Comments and Suggestions for Authors

Abstract:

  • Extended-spectrum β-lactamases (ESBL).

Response: Line 36: beta has been replaced with β.

  • Line 48: urinary tract infection (UTI)

Response: Line 50: Full for of UTI has been written.

  • Please explain the methodology in abstract shortly and clearly.

Response: Line 40-42: A short description regarding methodology has been added in the abstract.

  • Please amend the conclusion according to the result.

Response: Line 53-57: The conclusion section in the abstract has been amended.

Introduction:

  • This study is a fantastic one; thus, more papers should be used as strong references for this manuscript. Due to this fact, I recommend the authors to read and add the following papers to the References section of the manuscript to have fruitful Introduction and Discussion sections:

doi.org/10.1155/2022/5727638, doi.org/10.3390/pathogens11091019, doi.org/10.1128/spectrum.02137-22, doi: 10.2147/IDR.S356489, 10.1007/978-3-030-76320-6_20

Response: Dear reviewer, thank you for helping us by providing very nice studies to follow for introduction and discussion section. We have revised the introduction section and added these studies in the discussion section to strengthen the statements.

  • The introduction is too general: Report the epidemiology of carbapenemases and ESBLs that focus on the study area. Please show the study gaps.

Response: Line 66-85, 88-90, 92, 94-97, 103-108: Dear reviewer, thank you for your valuable suggestion. We have revised the introduction section, added the study gaps and improved the grammatical mistakes in the revised version of manuscript.

  • Line 80: I would like to write β-lactamases, not b-lactamases. Please amend this.

Response: Line 83: beta or b has been replaced with β.

  • Strong statements have been given in “Line 80-81” of the introduction with no references to support them. Please provide references

Response: Line 94-95: Dear reviewer, thank you for your valuable comment. We have cited the respective references to strengthen the statements.

  • Firstly, please write down the name of all bacteria into italic. Secondly, use the full name of bacteria in first i.e. Staphylococcus aureus(S. aureus) and then you can use the short name of bacteria i.e. S. aureus.

Response: Dear reviewer, we have corrected the bacterial names throughout the manuscript as per you comment and highlighted them in red colour.

  • Check reference number 7: Use an appropriate reference for nosocomial infection.

Response: Line 97: Dear reviewer, an appropriate reference has been cited to strengthen the statement.

Methodology

  • Line 201: Extended Spectrum β-lactamase (ESBLs)

Response: Line 251: The full form has been replaced with the abbreviation (ESBL).

  • Not clear regarding sample collection, transportation and preservation: How did you collect the blood sample? How did you transport the samples into microbiology lab? Did you preserve the sample? If yes, then for how many days and what was the temperature?

Response: Line 266-279: A description about sample collection has been written.

  • Section 4.5: Line 213, Cysteine Lactose Electrolyte Deficient agar?

Provide the manufacturer details of each chemical/material used (Company, City, Country).

Response: Line 280-289: A description about culture inoculation has been added and the company details for the agar media has been written in the revised manuscript. Furthermore, the full form for CLED has been corrected.

  • Why did inoculate urine samples on Blood and MacConkey agar? Please also give reference.

Response: Line 280-296: Dear reviewer, the urine samples were not inoculated on blood, chocolate and MAC agar plates. We have revised the related description.

  • “Positive samples were processed as per protocol and sample with no noticeable growth were ………………….. which it was declared negative” Not clear. Please modify this.

Response: Line 290-296: The description has been corrected.

  • Line 239: Enterobacteriaceae? Please write in italic.

Response: Line 298-299: Corrected.

  • Line 241: indole production? TSI agar for Hydrogen Sulphide? citrate usage? urease test? oxidase? carbohydrate utilization assays?

Response: Line 299-301: The sentence has been corrected.

  • Section 4.7: Clinical and Laboratory Standards Institute (CLSI)

Response: Line 303: Full for CLSI has been written.

  • In the method and result part, the authors used disk diffusion to do the antimicrobial activity. Is it about the measurement of zone inhibition or something else? This needs to be clarified.

Response: Line 315: Dear reviewer, the disk diffusion method was used as per the recommendation from CLSI guidelines. And after the incubation period, the zone of inhibitions was measured.

  • Section 4.7: Please give the reference why 0.5 MacFarland was used in AST?

Response: Line 306: Dear reviewer, a reference has been cited in the methodology. 0.5 MacFarland standard was used as per the recommendations from CLSI.

  • Section 4.7: Rephrase: The brackets are confusing, directly report antibiotic discs without classes or use table. Gentamicin (CN, 10μg)

Response: Line 310-314: The sentence has been corrected as per the suggestions.

  • Section 4.8: Screening of ESBLs by Phenotypic method: Please modify this heading. This section is not clear and creating confusion for reader. Please revise this accordingly what method was used for ESBLs identification?

Response: Line 320-323: Section 4.8 has been corrected as per the suggestion.

  • Section 4.9: Please write down the name of bacteria in italic (2ndparagraph).

Response: Line 345-346: The bacterial names has been italicized.

  • Section 4,10: Please write down the name of genes in italic (CTXblaNDMblaOXA-48, and blaKPCgenes sequences).

Response: Line 355: The gene names has been italicized throughout the manuscript.

  • Section 4.10.1: how did you confirm the DNA extraction either by spectrophotometer or gel electrophoresis?

Response: Line 361-365: The DNA confirmation was done by running the gel electrophoresis. A description has been written in the manuscript.

  • Did you preserve the DNA sample? If yes, what was SOPs followed?

Response: Line 361-362: A description has been added in section 4.10.1.

  • Section 4.10.3: Give references for the methods or provide details: What concentrations of dNTPs? Taq polymerase? Minerals? etc.   

Response: Line 385-388: The extra information has been removed and the reference studies has been cited.

  • Table 6: Please give the “Stands for” in legend. Tm? Please write the name of gene in italic.

Response: Table 6: The word TM has been removed from the table. The references have been provided in the text for the reader to refer to the parent study.

Results:

  • Section 2.1: What was the total number of samples among which 384 were positive for bacterial infection? Please mention.

Response: Line 111-114: The total number of samples has been added.

  • What about female? Please modify this accordingly.

Response: Line 114: The sentence has been corrected.

  • Table 1: ICU? OPD? Please mention legend.

Response: Line 122: The table legend has been written.

  • Section 2.2: what was the exact number of ESBL, CPR and CRE producing bacteria? Please mention it clearly and give along with % in Table 1.

Response: Table 3: The exact number of ESBL, CPR and CRE producing bacteria has been provided in table 3.

  • Table 4: What about ESBL and CRE producing bacteria? Please also mention in two separate tables.

Response: Dear reviewer, thank you for your valuable suggestion. Table 5 has been added in the revised manuscript to show the susceptibility pattern if ESBL-producing bacteria.

Discussion:

  • Revise the discussion accordingly. What did you expect?

Response: Line 162-168, 179-180, 208-235: The discussion section has been revised and more studies have been added to strengthen the discussion.

  • Avoid much repetitions of the results, please follow the STROBE guidelines

Response: The discussion section has carefully checked and repetitive results have been removed.

  • Please use the recent studies conducted in Pakistan and neighbouring countries to make fruitful Discussion.

Response: 208-235: More relevant studies have been added to strengthen the discussion.

  • Conclusion is not coming from Results. Please revise this. What about less resistant antibiotic either belong to carbapenem group or non-carbapenem group antibiotic? Could physician suggest less resistant antibiotics to treat infection? What was the conclusion of this study that will be beneficiary for Physician researcher and population? Please explain it clearly in conclusion.

Response: Line 402-406, 409-414: The conclusion section has been revised.

Reviewer 3 Report

My comments and suggestions

Line 58-59: A reference is missing

Line 62: Replace "carbapenem-resistant Enterobacterales" with "Carbapenem resistant Enterobacteriaceae"

Line 62-63: In my point of view, this sentence should be rewritten as for example: Carbapenems belong to the beta-lactam family. Antibiotics belonging to this family are widely used throughout the world to treat infectious diseases, with the exception of carbapenems. Antibiotics belonging to this sub-family of beta-lactam antibiotics are used as a last line of defence against multi-drug resistant bacteria, such as those producing ESBL. The current emergence of carbapenem resistance is a very serious public health problem.

Line 73: Replace "Microbe" with bacteria

Line 73-74: Polymyxin resistance remains very low in Gram- bacteria. It is better to rephrase this sentence by explaining the danger of the combination of carbapenem and polymyxin resistance.

Line 80-81: It is clear that carbapenemases hydrolyse carbapenems. I think this sentence should be rephrased as for example: The threat of CRE is mainly due to the emergence and spread of carbapenemase producing bacteria.

Line 87-88: Carbapenems belong to the beta-lactam family of antibiotics that inhibit the biosynthesis of plasma membrane peptidoglycan. So by what mechanism does sensitivity of bacteria to carbapenems lead to sensitivity to other families of antibiotics? I think this link is not correct.

Line 87-88: This sentence should be rewritten because bacteria that do not produce carbapenemases may be resistant or susceptible to other families of antibiotics. There is no link between carbapenem susceptibility and susceptibility to other families of antibiotics.

Line 89-90: "Thus, this carbapenem resistance may be associated with the presence or absence of carbapenemase-producing genes." This is a gratuitous and irrelevant statement .

Line 80 to 90: I do not understand this sentence. The author talks about the mechanism of resistance, then he talks about the probable origin of this resistance, then he comes back to the genetic mechanisms. I think we should first talk about mechanism (biochemical and genetic), then talk about the probable origin of these genes. I also think that carbapenem resistance does not only come from food. The increased consumption of carbapenems in human clinics could also increase the selection pressure of carbapenem resistance genes and lead to their appearance and maintenance.

Line 213: Replace 0.001 mm with 0.001 ml

Line 251-256: vharmonise the writing of antibiotic concentrations; group each antibiotic into its family; cefepime is 4th generation cephalosporin

Line 339: bla CTX is not a carbapenemase gene

the results should be presented taking into account the socio-demographic characteristics of the sampled patients

The discussion should also be made taking into account these socio-demographic factors. The prevalences of ESBL and CRE should be compared taking these factors into account. This would not only allow to know if the increase of ESBL is related to the emergence of CRE on the one hand; on the other hand, one should compare the prevalences of CRE in hospitalized and non-hospitalized patients at the level of the different hospital departments. This would give an idea of the origin of CRE (either consumption of carbapenems in hospital or contamination through food).

Author Response

Reviewer 3

Comments and Suggestions for Authors

My comments and suggestions

Line 58-59: A reference is missing

Response: Line 69: References has been cited to strengthen the statements.

Line 62: Replace "carbapenem-resistant Enterobacterales" with "Carbapenem resistant Enterobacteriaceae"

Response: Line 67: Enterobacterales has been replaced with Enterobacteriaceae.

Line 62-63: In my point of view, this sentence should be rewritten as for example: Carbapenems belong to the beta-lactam family. Antibiotics belonging to this family are widely used throughout the world to treat infectious diseases, with the exception of carbapenems. Antibiotics belonging to this sub-family of beta-lactam antibiotics are used as a last line of defence against multi-drug resistant bacteria, such as those producing ESBL. The current emergence of carbapenem resistance is a very serious public health problem.

Response: Line 69-71: The sentence has been rewritten.

Line 73: Replace "Microbe" with bacteria

Response: Line 87: Microbes has been replaced with bacteria.

Line 73-74: Polymyxin resistance remains very low in Gram- bacteria. It is better to rephrase this sentence by explaining the danger of the combination of carbapenem and polymyxin resistance.

Response: Line 88-89: The sentence has been rephrased and corrected.

Line 80-81: It is clear that carbapenemases hydrolyse carbapenems. I think this sentence should be rephrased as for example: The threat of CRE is mainly due to the emergence and spread of carbapenemase producing bacteria.

Response: Line 84-85: The sentence has been rephrased.

Line 87-88: Carbapenems belong to the beta-lactam family of antibiotics that inhibit the biosynthesis of plasma membrane peptidoglycan. So by what mechanism does sensitivity of bacteria to carbapenems lead to sensitivity to other families of antibiotics? I think this link is not correct.

Line 87-88: This sentence should be rewritten because bacteria that do not produce carbapenemases may be resistant or susceptible to other families of antibiotics. There is no link between carbapenem susceptibility and susceptibility to other families of antibiotics.

Response: Line 94-95: Dear reviewer, thank you for your valuable comment. The sentence has been rephased and corrected.

Line 89-90: "Thus, this carbapenem resistance may be associated with the presence or absence of carbapenemase-producing genes." This is a gratuitous and irrelevant statement.

Response: The sentence has been removed.

Line 80 to 90: I do not understand this sentence. The author talks about the mechanism of resistance, then he talks about the probable origin of this resistance, then he comes back to the genetic mechanisms. I think we should first talk about mechanism (biochemical and genetic), then talk about the probable origin of these genes. I also think that carbapenem resistance does not only come from food. The increased consumption of carbapenems in human clinics could also increase the selection pressure of carbapenem resistance genes and lead to their appearance and maintenance.

Response: Line 94-97: The paragraph structure has been corrected. Furthermore, Line 67-82, 85-88, 90, 92-95, 101-106: We have revised the introduction section, added the study gaps and improved the grammatical mistakes in the revised version of manuscript.

Line 213: Replace 0.001 mm with 0.001 ml.

Response: Dear reviewer, thank you for your valuable suggestion. We have removed the estimated quantity volume. As suggested by another reviewer.

Line 251-256: vharmonise the writing of antibiotic concentrations; group each antibiotic into its family; cefepime is 4th generation cephalosporin.

Response: Line 310-314: The names for antibiotics disks has been corrected.

Line 339: bla CTX is not a carbapenemase gene

Response: Line 389: The mistakes has been corrected.

the results should be presented taking into account the socio-demographic characteristics of the sampled patients.

Response: Dear reviewer, thank you for your valuable suggestion which have improved the quality and representation of manuscript. Table 2 and Table 3 has been amended, and more data has been added.

The discussion should also be made taking into account these socio-demographic factors. The prevalences of ESBL and CRE should be compared taking these factors into account. This would not only allow to know if the increase of ESBL is related to the emergence of CRE on the one hand; on the other hand, one should compare the prevalences of CRE in hospitalized and non-hospitalized patients at the level of the different hospital departments. This would give an idea of the origin of CRE (either consumption of carbapenems in hospital or contamination through food).

Response: Line 162-168, 179-180, 208-235: The discussion section has been revised and more studies have been added to strengthen the discussion.

Round 2

Reviewer 1 Report

I approve the edits by the authors. They have improved their data analysis, and cleaned up the discussion and conclusions sections.

Author Response

Reviewer 1

Comments and Suggestions for Authors

I approve the edits by the authors. They have improved their data analysis, and cleaned up the discussion and conclusions sections.

Author response: Dear reviewer, thank you for approving our efforts and suggesting the manuscript for the possible publication process. After addressing the comments from you and other reviewers, the manuscript quality has been significantly improved.

Thank you!

Reviewer 2 Report

The authors have addressed my suggestions/comments and the revised manuscript looks good.

The authors should do a few minor changes.

Please replace Enterobacterales with Enterobacteriaceae.

Table 3: E. coli, ICU. Please use 6, 16.1, not 6, 16.14 (Single digit after the dot). please amend all tables, accordingly.

Author Response

Reviewer 2

Comments and Suggestions for Authors

The authors have addressed my suggestions/comments and the revised manuscript looks good.

The authors should do a few minor changes.

Author response: Dear reviewer, thank you for approving our efforts and suggesting the manuscript for the possible publication process. After addressing the comments from you and other reviewers, the manuscript quality has been significantly improved.

Please replace Enterobacterales with Enterobacteriaceae.

Author response: We have revised “Enterobacterales” with “Enterobacteriaceae” throughout the manuscript.

Table 3: E. coli, ICU. Please use 6, 16.1, not 6, 16.14 (Single digit after the dot). please amend all tables, accordingly.

Author response: Dear reviewer, thank you for your valuable suggestion. We have revised the percentages throughout the manuscript.

Reviewer 3 Report

For me this article can be published after some corrections and clarifications:

 Line 78-79 : The link between ESBLs and Carbapenem is not well understood. I think it would be better to rewrite this sentence as follows : The extended-spectrum β-lactamases (ESBLs) have been seen as a serious threat to 77 public health since the beginning of the century [6]. Carbapenems antibiotics belong Betalactam familly.  ESBLs can hydrolyse the majority of antibiotics belonging to the β-lactam family, except for carbapenems

Line 85-86 : Carbapenems belong to the β-lactam family. Antibiotics belonging to this family are  widely used throughout the world to treat infectious diseases : should be deleted

 Line 86-87 : Carbapenem antibiotics are used as a last line of defense against multi-drug resistant (MDR) bacteria

Line 101-112 : Could be improved as follows: …….Enterobacteriaceae [20]. A recent study from Punjab, Pakistan has re-100 ported 14.4% CP producing Enterobacteriaceae [20]. The increased consumption of antibiotics in clinical settings could also increase the  selection pressure of antibiotic resistance genes and may lead to high antimicrobial re-108 sistance rates [6,25]. However, the growth of carbapenem resistance could also be caused by the consumption of fishery products (ref). There have been several reports of bacteria recovered from fisheries products with plasmids carrying genes for carbapenem resistance [21-23]. Commercial shrimp were found to include Vibrio alginolyticus carrying the encoding VIM-1 and the encoding NDM-1 genes in a Chinese retail market [24]. In Vitenam, Enterobacteriaceae carrying the VIM-1 and NDM-1,4,5, KPC and OXA-48 have 105 been isolated from fish and other sea animals [23]. In Pakistan, the antimicrobial resistance rates are continuously in creasing and leading to significant health threats [2,8]. The current study was conducted with the aims to see the prevalence of CP, CRE and ESBL producing Enterobacteriaceae both phenotypically and genotypically.

Line 114-162: This would make more sense than the results presented as follows:

2.1 demographic characteristics

2.2 Antibiotic resistance patterns

2.3 ESBL and Carbapenemase production profiles

2.4 Real-time multiplex PCR analysis

line 208-210: This sentence is not clear. drainage system of what?

There is no link between this drainage system and the rate of carbapenem resistance. You would have to explain how city dwellers have higher carbapenem resistance rates than rural inhabitants

Line 212: Replace thr with the

Author Response

Reviewer 3

Comments and Suggestions for Authors

For me this article can be published after some corrections and clarifications:

Line 78-79 : The link between ESBLs and Carbapenem is not well understood. I think it would be better to rewrite this sentence as follows : The extended-spectrum β-lactamases (ESBLs) have been seen as a serious threat to 77 public health since the beginning of the century [6]. Carbapenems antibiotics belong Betalactam familly.  ESBLs can hydrolyse the majority of antibiotics belonging to the β-lactam family, except for carbapenems

Author response: Line 77-80: Dear reviewer, thank you for your valuable suggestion. We have revised the sentence as per your suggestion.

Line 85-86 : Carbapenems belong to the β-lactam family. Antibiotics belonging to this family are  widely used throughout the world to treat infectious diseases : should be deleted

Author response: The sentence has been removed from the revised manuscript.

Line 86-87 : Carbapenem antibiotics are used as a last line of defense against multi-drug resistant (MDR) bacteria

Author response: Line 86: The sentence has been revised.

Line 101-112 : Could be improved as follows: …….Enterobacteriaceae [20]A recent study from Punjab, Pakistan has re-100 ported 14.4% CP producing Enterobacteriaceae [20]. The increased consumption of antibiotics in clinical settings could also increase the  selection pressure of antibiotic resistance genes and may lead to high antimicrobial re-108 sistance rates [6,25]. However, the growth of carbapenem resistance could also be caused by the consumption of fishery products (ref). There have been several reports of bacteria recovered from fisheries products with plasmids carrying genes for carbapenem resistance [21-23]. Commercial shrimp were found to include Vibrio alginolyticus carrying the encoding VIM-1 and the encoding NDM-1 genes in a Chinese retail market [24]. In Vitenam, Enterobacteriaceae carrying the VIM-1 and NDM-1,4,5, KPC and OXA-48 have 105 been isolated from fish and other sea animals [23]. In Pakistan, the antimicrobial resistance rates are continuously in creasing and leading to significant health threats [2,8]. The current study was conducted with the aims to see the prevalence of CP, CRE and ESBL producing Enterobacteriaceae both phenotypically and genotypically.

Author response: Dear reviewer, thank you for your valuable suggestions. We have revised the paragraph as per your suggestion.

Line 114-162: This would make more sense than the results presented as follows:

2.1 demographic characteristics

2.2 Antibiotic resistance patterns

2.3 ESBL and Carbapenemase production profiles

2.4 Real-time multiplex PCR analysis

Author response: Dear reviewer, thank you for your valuable suggestions. We have revised the results section in the revised manuscript accordingly.

line 208-210: This sentence is not clear. drainage system of what?

There is no link between this drainage system and the rate of carbapenem resistance. You would have to explain how city dwellers have higher carbapenem resistance rates than rural inhabitants

Author response: Dear reviewer, thank you for highlighting the point and we apologies for the mistake as well. We have removed this point from the revised manuscript as in the current study, there was no drainage samples.

Line 212: Replace thr with the

Author response: Line 214: Spelling has been corrected.